# Graphene Oxide-Induced Protein Conformational Change in Nasopharyngeal Carcinoma Cells: A Joint Research on Cytotoxicity and Photon Therapy

**DOI:** 10.3390/ma14061396

**Published:** 2021-03-13

**Authors:** Selvaraj Rajesh Kumar, Ya-Hui Hsu, Truong Thi Tuong Vi, Jong-Hwei Su Pang, Yao-Chang Lee, Chia-Hsun Hsieh, Shingjiang Jessie Lue

**Affiliations:** 1Department of Chemical and Materials Engineering, Chang Gung University, Wenhua 1st Road, Guishan, Taoyuan 333, Taiwan; rajeshkumarnst@gmail.com (S.R.K.); truongthituongvi005@gmail.com (T.T.T.V.); 2Graduate Institute of Clinical Medical Sciences, Chang Gung University, Wenhua 1st Road, Guishan, Taoyuan 333, Taiwan; hyh17@cgmh.org.tw (Y.-H.H.); jonghwei@mail.cgu.edu.tw (J.-H.S.P.); 3Department of Physical Medicine and Rehabilitation, Chang Gung Memorial Hospital, Dinghu Road, Guishan, Taoyuan 333, Taiwan; 4National Synchrotron Radiation Research Center, Hsin Ann Road, Hsinchu City 300, Taiwan; yclee@nsrrc.org.tw; 5Division of Hematology-Oncology, Department of Internal Medicine, New Taipei Municipal TuCheng Hospital, Jincheng Road, New Taipei City 236, Taiwan; 6Division of Hematology-Oncology, Department of Internal Medicine, Chang Gung Memorial Hospital at Linkou, Fusing Street, Guishan, Taoyuan 333, Taiwan; 7School of Medicine, Chang Gung University, Wenhua 1st Road, Guishan, Taoyuan 333, Taiwan; 8Division of Join Reconstruction, Department of Orthopedics, Chang Gung Medical Center at Linkou, Fusing Street, Guishan, Taoyuan 333, Taiwan; 9Department of Safety, Health and Environment Engineering, Ming-Chi University of Technology, Gongzhuan Road, Taishan, New Taipei City 243, Taiwan; 10Center for Environmental Sustainability and Human Health, Ming-Chi University of Technology, Gongzhuan Road, Taishan, New Taipei City 243, Taiwan

**Keywords:** graphene oxide nanosheets, synchrotron radiation, cell viability, cancer cells, photon therapy

## Abstract

The objectives of this work aim to investigate the interaction and cytotoxicity between nanometric graphene oxide (GO) and nasopharyngeal carcinoma cells (NPC-BM1), and possible application in photon therapy. GO nanosheets were obtained in the size range of 100–200 nm, with a negative surface charge. This nanometric GO exhibited a limited (<10%) cytotoxicity effect and no significant dimensional change on NPC-BM1 cells in the tested GO concentration range (0.1–10 µg·mL^−1^). However, the secondary protein structure was modified in the GO-treated NPC-BM1 cells, as determined through synchrotron radiation-based Fourier transform infrared microspectroscopy (SR-FTIRM) mapping. To further study the cellular response of GO-treated NPC-BM1 cancer cells at low GO concentration (0.1 µg·mL^−1^), photon radiation was applied with increasing doses, ranging from 2 to 8 Gy. The low radiation energy (<5 Gy) did not cause significant cell mortality (5–7%). Increasing the radiation energy to 6–8 Gy accelerated cell apoptosis rate, especially in the GO-treated NPC-BM1 cells (27%). This necrosis may be due to GO-induced conformational changes in protein and DNA/RNA, resulting in cell vulnerability under photon radiation. The findings of the present work demonstrate the potential biological applicability of nanometric GO in different areas, such as targeted drug delivery, cellular imaging, and radiotherapy, etc.

## 1. Introduction

In recent years, multimodal noninvasive clinical imaging and radiological treatments have been established to achieve an accurate range of diagnostic targeted areas and therapeutic procedures for critical diseases. Photon radiotherapy is an important part of cancer treatment that is received by approximately two-thirds of cancer patients (50–60%) [1]. Recently, the use of nanomaterials has led to enhanced penetration ability of the substances employed for photon radiotherapy and diagnosis with lower risk than those of conventional drugs [2]. Therefore, the ultimate aim of photon radiotherapy is to deliver a prescribed dose to a tumor precisely while minimizing side effects to the surrounding tissues.

Graphene oxide (GO) is a fascinating atomic sheet of exfoliated oxidized graphite. It contains oxygen functional groups on its edges and/or basal planes to form a hybrid structure that has σ states and π state from the sp^3^-hybridized and sp^2^-hybridized carbon domains. The GO toxicity effects are closely dependent on the surface structure, lateral size, surface functionalization, surface charge, and propensity to adsorb proteins [3]. Moreover, the GO cytotoxicity mechanisms involve different pathways, including impairment of mitochondrial activity, plasma membrane damage, DNA damage and induction of oxidative stress [4,5]. Thereby, the graphene-related nanomaterials have been widely investigated for their antineoplastic effect on cellular toxicity toward various cancer cell lines (head and neck, ovarian, breast, lungs, etc.) [6]. For cancer treatments, graphene or GO can induce cell death by various mechanisms: wrapping of cells, free radical generation, nuclear damage, autophagy, and apoptosis [7,8]. Accordingly, GO or graphene internalization leads to toll-like receptor expression, cell shrinkage [9], and changes in gene expression [10] that could cause autophagy and anticancer activities. However, some reports have revealed that the internalization of graphene or GO might transport therapeutic agents intracellularly without subsequent cancer cell damage [11]. For example, GO particles are less toxic to various cancer cells than to normal fibroblastic cells due to different administration routes, and physiological barriers [12,13]. Thereby, the study of GO functional group changes with cancer cell interactions is important for targeted drug delivery applications.

Among several cancer types, nasopharyngeal carcinoma (NPC) is also highly sensitive to radiotherapy and is mostly diagnosed at an advanced stage [14]. The five-year overall survival rate of stage III NPC patients receiving chemoradiation is >72% [8]. Therefore, targeting NPC cells with suitable drug carriers is an important factor for chemotherapy and radiosensitive survival. The GO nanocarrier can effortlessly deliver anticancer drugs or biomolecules (peptides, proteins, nucleic acids, etc.) into targeted cancer cells through the cell membrane [9]. Moreover, the cumulative drug release rate from GO nanocarriers depends on π-π stacking and hydrogen bonding interactions between GO and the drug molecules [15,16]. Many studies have successfully demonstrated targeted drug delivery to cancer cells using GO nanocarriers [17,18]. Recently, Lan et al. [14] reported suppression of NPC cell proliferation, invasion, and migration when using polyethylene glycol coated GO loaded with erlotinib particles and showed potential anticancer activities compared with free drug molecules. Conversely, data about the GO functional group changes and its interaction with NPC cancer cells are limited in the literature and increasing this knowledge is a significant scientific quest that needs to be addressed.

Among many techniques, FTIR microspectroscopy (FTIRM) mapping is used for studying functional group and biochemically induced changes in the single or multiple cells using highly sensitive synchrotron radiation (SR) facilities [19,20,21]. FTIRM is a label-free analytical tool and non-destructive approach that provides more reliable insights into the spectral signatures and a spatial location of the chemical components to identify the cellular alternations and their functions including lipids, proteins, and nucleic acids [22,23]. For example, Jamin et al. demonstrated the functional groups of biomolecules including nucleic acids, lipids, and proteins in a single live cell using synchrotron radiation-based (SR)-FTIRM mapping with a spatial resolution of a few microns [24]. Moreover, the earlier stage of pathological variations in the cancer cells is also clearly determined by SR-FTIRM [25]. Mahmoud et al. represented that FTIRM was a unique technique for the detection of malignant cells when compared with consistent difference between normal cells [26]. Domenici et al., [27] recognized the single-cell level of spectral fingerprints, functional groups, and biochemical changes of gold nanoparticles-treated fibroblast cells through SR-FTIRM. Recently, Vongsvivut et al. [19] used SR-FTIRM to demonstrate the mapping for individual neuron in brain tissue, single cell of malaria-infected red blood cells (RBC) and *Eucalyptus* leaf cell structure. The result confirms that the single RBC was infected by the malarial pathogen of Plasmodium parasite, chemical composition of neutrons, and the epicuticular wax formation and the plant lipid metabolism. Therefore, the SR-FTIRM is an excellent technique for biomolecule analysis that provides the spectral phenotypes and detailed contribution of all the macromolecules reflected in the single cell.

The objective of the present work was to investigate the cytotoxicity and elucidate the mechanism of cellular interaction between NPC-BM1 cells and nanometric GO and to study the photon radiation effects of GO (Figure 1). To achieve this goal, nanometric GO was prepared through a modified Hummers method and probe-sonication process. The physicochemical and morphological properties of nanometric GO are discussed in detail, confirming the success of graphite exfoliation. Then, the cytotoxic effect of nanometric GO-treated NPC-BM1 cells was studied at various time intervals. The macromolecules present in the pure NPC-BM1 and nanometric GO-treated NPC-BM1 cells were studied using SR-FTIRM mapping. Furthermore, the radiation effect and anticancer activity on GO-treated NPC-BM1 cells were investigated with different photon energies.

## 2. Materials and Methods

### 2.1. Preparation of Nanometric GO

Based on our previous studies, GO was synthesized using a modified Hummer’s technique [28,29]. The detailed procedure for GO synthesis was provided in the Appendix A. The nanometric GO was prepared by the probe sonication method [30]. The prepared GO was then dispersed in DI water to obtain a GO colloidal solution and probe-sonicated for 120 min via ultra-probe sonicator (Qsonica, Sunway Scientific Corporation, Hsinchu, Taiwan). During probe sonication, an ice bath was employed to avoid increasing the temperature. Finally, the sample was dialyzed using a dialysis bag for 24 h and dried in a vacuum oven. The final product of 100–200 nm sized GO was kept in a dry, cool area for further analysis. Ten milligrams of the prepared nanometric GO were dissolved in 100 mL DI water to obtain homogeneous GO suspension by ultrasonication to form a stock solution.

### 2.2. Characterization

The morphology of the graphene and GO was viewed using a transmission electron microscope (TEM, JEM-2000EXII, JEOL, Tokyo, Japan). The phase purity and crystal structure of graphite and GO were evaluated by X-ray diffraction (XRD, model D5005D, Siemens AG, Munich, Germany). Fourier transform infrared spectroscopy (FTIR, model Horiba FT-730, Minami-ku, Kyoto, Japan) was employed to investigate the chemical structure of graphene and GO. Furthermore, the detailed structural properties of carbon-containing functional groups were studied using micro-Raman spectroscopy (LabRam HR 800, Horiba, Ltd., Kyoto, Japan). The chemical composition of GO was examined by X-ray photoelectron spectroscopy (XPS K-Alpha, VG Microtech MT-500, Thermo Fisher Scientific Inc., Waltham, MA, USA). The GO surface charge and particle size distribution were determined from the dynamic light scattering (DLS, Zetasizer, 2000 HAS, Malvern, Worcestershire, UK) at ambient temperature.

### 2.3. Sample Preparation and Synchrotron-Based FTIR Mapping of Cells

Initially, the GO-treated nasopharyngeal cancer cells (NPC-BM1, HELIX Technology Co. Ltd., Guishan District, Taoyuan City, Taiwan) (1 × 10^6^ cells/mL) were seeded onto conductive Ag/SnO_2_-coated IR reflective low-e microscopic slides (Kevley Technologies, Chesterfield, OH, USA) and allowed to grow for 24 h. Afterwards, the culture medium was removed, and the cells were washed using phosphate buffer solution (PBS, Sigma-Aldrich, St. Louis, MO, USA) at ambient temperature. Then, 4% paraformaldehyde was added to fix the cells at 4 °C for 30 min. The cell-seeded slides were finally washed two times with PBS and DI water. Finally, the slides were dried and stored for further analysis. FTIR mapping was performed by using a synchrotron-radiation-based Fourier transform infrared microspectroscopy (SR-FTIRM), which includes an FTIR spectrometer (Nicolet 6700, Thermo Fisher Scientific, Madison, WI, USA) and a confocal infrared microscope (Nicolet Continuum; Thermo Fisher Scientific, Madison, WI, USA) from the National Synchrotron Radiation Research Center (NSRRC), Hsinchu, Taiwan, at the TLS 14A1 infrared microspectroscopy (IMS) endstation. A total of 128 scans of FTIR spectra of a single cell at 4 cm^−1^ resolutions in the spectral range of 4000–650 cm^−1^ were acquired using synchrotron IR radiation. The final FTIR mapping was observed through a confocal aperture.

### 2.4. Cell Viability Assay

Initially, NPC-BM1 cells were cultured at 37 °C in 5% CO_2_ with 95% relative humidity for 24 h. Then, 1 × 10^5^ cells/mL (cell density) were placed in 24-well plates. Different GO concentrations (0.1, 0.5, 1, 5 and 10 µg mL^−1^, which were diluted 1, 5, 10, 50, and 100 µL·mL^−1^ from GO stock solution) were added to the NPC-BM1 cells. Again, the samples were placed in an incubator at 37 °C in 5% CO_2_ with 95% relative humidity for 24, 48, and 72 h (time-dependent experiments). Afterwards, 50 mg mL^−1^ 3-(4,5-dimethylthiazol-2-yl)-2,5-diphenyltetrazolium bromide (MTT) agent was added to each sample and the cells were again incubated for 1 h under similar conditions. The optical density (OD) at a wavelength of 595 nm of the GO-treated cells was analyzed via UV-Vis spectroscopy. The microscopy images of GO-treated cells were analyzed at a magnification of 100×. All the experiments were reported as the average value of measurements from three replications (*n* = 3).

### 2.5. Radiation Treatment

Photon radiation of GO-treated NPC-BM1 cells was performed at Chang Gung Memorial Hospital, Linkou, Taiwan. The NPC-BM1 cells were seeded into 6-well plates (1 × 10^5^ cells mL^−1^) and incubated overnight. Subsequently, the cells were cultured with cell medium containing 0.1 μg·mL^−1^ GO for 24 h. The distinctly stained cells were exposed to X-ray radiation at various doses (2, 3, 5, 6, and 8 Gy). After photon irradiation, the cell viability was again determined by using the MTT assay. To obtain consistent data, all the sample analyses were repeated in triplicate (*n* = 3), and the average values were reported.

## 3. Results and Discussion

### 3.1. Surface Morphology and Structural Analysis

The TEM micrograph of graphite revealed a large, wrinkled sheet-like structure with a number of layers, and the size ranged from 10–20 µm, as displayed in Figure 2a. This wavy wrinkled shape fragmented into smaller GO particles with a single-layered structure in the size range of 100–200 nm, as shown in Figure 2b. The reduction in GO particle size was due to the breaking of graphene sheets through the exfoliation process via a strong probe-sonication process [31]. The average GO particle size distribution ranged from 50 nm to 200 nm determined by dynamic scattering light, as shown in Appendix A.

The phase purity of graphite and GO was studied using XRD patterns (Figure 3a). The sharp diffraction peak at 26.2° was attributed to the (002) plane of the graphite structure with a d-spacing of 3.4 Å. After oxidation, the diffraction peak at 11° was attributed to the (011) plane of GO with an increased d-spacing of 7.94 Å. This was due to the hydrophilic nature of oxygen-containing groups, including carboxyl and epoxy groups in the GO basal plane throughout the oxidation routes [28].

The functional groups of the graphite and GO were investigated using FTIR spectroscopy, as shown in Figure 3b. The graphite FTIR peaks did not indicate noteworthy oxygen-containing functional groups [32,33], whereas several representative peaks were observed for the GO particles. The FTIR peaks at 1218 cm^−1^ and 1051 cm^−1^ were attributed to the stretching vibration of C–O in the epoxy alkyl group. The sharp FTIR peaks at 1627 cm^−1^ and 1732 cm^−1^ corresponded to the aromatic vibration of the C=C bonds and the C=O stretching vibration of GO. The corresponding peak at 3390 cm^−1^ was attributed to the stretching and deformation vibrations of the surface hydroxyl molecules present in GO. All the FTIR peaks established successful GO formation through graphite exfoliation [28,34].

Raman spectroscopy was further used to examine the chemical structure of the carbon-based nanomaterials and their ID/IG ratio (Figure 3c). For graphite, the strong peak at 1574 cm^−1^ (G band) represents the first-order scattering of the E2g mode, and the small band at 1352 cm^−1^ (D band) provides evidence for the presence of defects in the graphite. After exfoliation, the GO spectrum of the G band was shifted towards higher wavenumbers (1584 cm^−1^) owing to graphite oxidation. Additionally, the increased peak intensity of the D band indicated the formation of disorder/defects, including the presence of aliphatic chains, grain boundaries, and in-plane heteroatoms, and confirmed the oxidation of graphite owing to structural changes [31]. The ID/IG intensity ratio was 0.16 for graphite, whereas the ratio increased to 0.96 for the GO sample. The 2D peak of graphite was observed at ~2694 cm^−1^. After exfoliation from graphite, the 2D peak intensity was significantly suppressed and marginally shifted through disruption of the stacking order owing to the oxidation reaction [35]. This result signifies the formation of a monolayer of GO nanosheets.

The chemical composition of GO was estimated using the XPS spectrum, and the full scan data are shown in Figure 3d. The chemical composition of graphite includes 2% oxygen and 98% carbon, as reported in previous work [28]. After exfoliation, the GO oxygen content significantly increased to 32%, and the carbon content decreased to 68%, as calculated from deconvolution of the peaks. The higher GO oxygen content further confirms the existence of more hydrophilic groups [36]. The surface charge of nanometric GO was calculated as -38 mV due to the ionization of carboxylic acid groups and the formation of a stable aqueous dispersion [28]. In our previous report, the prepared GO exhibited a negative surface charge (from −35.7 to −22.4 mV) throughout a wide pH range (pH of 11.5–2.4) [37].

### 3.2. Cell Cytotoxicity

The cytotoxicity of control and nanometric GO-treated NPC-BM1 cells was studied in a dose-dependent manner, and the data were reported as viable cell percentages. The GO concentration-dependent cytotoxic effects (0.1–10 µg mL^−1^) of NPC-BM1 cells at different time periods (24, 48, and 72 h) are presented in Figure 4. No noteworthy reduction in their metabolic activity at first 24 h was observed after exposure to the lower GO concentration (0.1–0.5 µg mL^−1^) compared with the control. Increasing the GO concentration (1–10 µg·mL^−1^) caused a slight reduction in cell viability.

Upon increasing the incubation period to 48 h and 72 h, the viability of cells was similar to the 24 h incubation results for lower (0.1–0.5 µg·mL^−1^) and higher GO concentrations (1–10 µg·mL^−1^). This was similar to the results of Wu et al. [38], in which no cytotoxicity effects of GO-treated multiple myeloma cells were observed and GO did not induce cell apoptosis in RPMI-822 cells. Moreover, Zhang et al. [39] reported that smaller GO particles caused less damage to the cell membrane of cancer cells through a physical damage mechanism than larger GO nanosheets (>1 µm). At higher GO concentrations, the aggregation effects might change the microenvironment of the NPC-BM1 cells to induce cytotoxicity effects. A similar range of cytotoxicity effects for different cancer cells with high GO concentrations have been reported in the literature [6,13]. Therefore, GO could possibly serve as a stable nanocarrier upon different time periods for targeted drug delivery and efficient radiotherapy routines.

Microscopic images of the pristine and various GO-treated cells are shown in Figure 5. Figure 5c shows the decreased NPC-BM1 cell density (as indicated by the arrow mark) due to GO interaction and its agglomeration effects when compared with the control (Figure 5a) and lower GO concentration (Figure 5b). Similar findings were also reported that a higher concentration of carbon- or GO-based nanomaterials induced milder toxicity effects than those induced by lower concentration levels [5,40]. Recently, Hara et al. [41] reported that GO particle concentration exceeding 10 µM leads to cell cytotoxicity. This was due to the presence of more hydrocarbon groups participating in intramolecular interactions with the nitrogen groups of targeted cells leading to mildly increased cytotoxicity. Likewise, the nanometric GO has more hydrocarbon functional groups that directly interact with the amide group of NPC-BM1 cells to influence mild cell toxicity at higher concentrations. In another possible mechanism, GO at a higher concentration could wrap the cell membrane to restrict cell replication [42] and may induce mild cell toxicity without GO particle internalization into NPC-BM1 cells. Based on the present results, GO is a highly biocompatible and suitable nanocarrier for targeted drug delivery.

### 3.3. FTIR Mapping of GO-Treated NPC-BM1 Cells

FTIR spectroscopy, electron microscopy imaging, and 2D FTIRM mapping were further used to study the morphology and functional group changes of the NPC-BM1 cells, as displayed in Figure 6. All the corresponding FTIR absorption bands (Figure 6a) were closely matched with those of functional groups of fibroblasts, yeast, CaCo-2, HeLa, PC3, and endothelial cells reported in the literature [27,43,44,45]. Figure 6a shows that there were significant peak intensity increases in the NPC-BM1 cells after the GO treatment, particularly in the amide I and amide II functional groups (at 1644 and 1536 cm^−1^, respectively). These amide groups are related to protein components [46,47,48]. The amide I groups consists of C=O stretching in protein molecules, whereas amide II represents the δN–H bending and vC–N stretching vibrations [43]. Bassan et al. [20] reported the increased amide II band intensity (at 1535–1545 cm^−1^) for renal tumor cell when compared with normal tissue. Holman et al. [46] normalized IR spectra (of normal human fetal lung fibroblast) to amide I peak intensity, and discovered that the amide II band increased, along with peak wavelength shifting downward, for dying cell as opposed to normal living cell. They ascribed the peak shift to lower wavelength to the protein structural change from α-helix to β-sheet [46,48,49]. Hence, both amide I and amide II peak shift and their intensity variation in GO-treated sample confirms the change in the secondary structure of proteins, as compared with pristine NPC-BM1 cells.

We examined the morphology of the cells before and after GO treatment and the typical images are shown in Figure 6b,f. The equivalent diameter of the pristine NPC was 37.9 ± 3.4 μm and that of the GO-treated one was 38.0 ± 3.7 μm, without statistically significant difference (*p*-value > 0.05) between the two groups. This indicated no visible dimensional change when the cells were exposed to the low GO dose (0.1 µg·mL^−1^).

The FTIR mapping was further employed to examine the protein conformational changes within the cells prior to and post GO treatment. Since amide I absorption resulted in the most dominant peak in the FTIR spectra (Figure 6a), its intensity was adopted to map the cells and color spectrum was assigned according to the intensity contour of this functional group. The amide I intensified near the center of the cells and faded toward the cell border (Figure 6c,g). At an intensity threshold of 0.25 and labelled with bright turquoise color, this boundary may be referred as the cell boundary. These regions had equivalent diameters of 26.7 and 25.9 μm for pristine and GO-treated NPC cells, respectively. The major amide I amplitude absorption was taken at an intensity threshold of 0.65 and presented as red regions in Figure 6c,g. We estimated that the equivalent diameters of these regions were 7.2 and 10 μm for pristine and GO-treated NPC cells, respectively. The peak intensity of amide I (1644 cm^−1^) increased in GO-treated cancer cells due to modification in protein content as compared with a control group. This was probably owing to the GO interaction with NPC-BM1 cells and its functional group changes. The oxygen functional groups in GO may induce C=O bond formation with amino acids in protein or nitrogenous bases in DNA/RNA.

The FTIR peak intensity of the amide II band was also increased, and the peak shifted to a lower wavenumber (1518 cm^−1^) due to the presence of GO (Figure 6a). The changes in the intensity of the amide II bands were attributed to the structural alterations in proteins within cells [46,47,50]. This fact was in line with the increased area of the red region from amide II (Figure 6d,h), indicating changes in their protein secondary structure, from α-helix to β-sheet, near the cell center [44]. Moreover, the amide I/amide II absorbance ratio also indicates some overall protein conformational states within the cancer cell [46] due to nanometric GO internalization. The FTIR band at 1234 cm^−1^ represents asymmetric stretching vibration of phosphate (ν_as_PO_2_^−^) head groups of phospholipids [51]. This peak shifted to a lower wavenumber (1227 cm^−1^) in the GO-treated cells (Figure 6a), which arises from the asymmetric vibration of the phosphate group in the NPC-BM1 cell. The corresponding SR-FTIRM mapping images of pristine and GO-treated NPC-BM1 cells are shown in Figure 6e,i. The red regions, representing phosphate-rich regions, of the GO-treated NPC-cells were larger than that of the pristine sample. Remarkably, the band at 1062 cm^−1^ arises from the symmetric PO_2_ (ν_s_PO_2_^−^) stretching vibrations of the phosphate groups in the DNA double-stranded backbone. The peak shift to the lower wavenumber of these two adsorption bands (1234 and 1062 cm^−1^) was reported to reflect the cell DNA/RNA content, which indicated the degree of oxidative damage [43,47]. Roveria et al. [48] represented that the AuNPs and GdNPs-treated F98 glioma cells show lower wavenumber shift in ν_as_PO_2_^−^ and indicate conformation changes in the DNA backbone owing to nucleotide base damage from the partial transition between B-DNA to A-DNA structures. In the present work, the GO may damage nucleotide and phospholipid groups of NPC-BM1 cells because the negative surface charge of nanometric GO could induce a strong electrostatic interaction with the positively charged lipids [5] present on the NPC-BM1 cells. Thereby, the nanometric GO could undergo cell wrapping and then penetrate the intracellular compartment, resulting in disruption of the phospholipid bilayer [4] through endocytic mechanisms [52] and increase DNA oxidative stress [5,53]. Overall, upon exposure to the low dose of GO (0.1 µg·mL^−1^), the NPC cell size was not altered but the conformational modifications on the protein and DNA/RNA were observed.

### 3.4. Photon Radiation Effect on NPC-BM1 Cancer Cell

Dose-dependent photon radiation effect on GO-treated NPC-BM1 cells (GO: 0.1 µg·mL^−1^) was performed and compared with the control group (without GO-treated), as shown in Figure 7. Without radiation (i.e., at zero dosage), the cell survival rate of the NPC-BM1 cell exposed to GO was the same as that without exposure, indicating that this nanometric GO was not harmful to the cell at this low GO concentration. When exposed to radiation doses of 2–5 Gy, the GO-treated NPC-BM1 cells did not experience severe cell apoptosis and displayed similar cell compatibility of 93–95% (with GO treatment) and 89–99% (without GO treatment). These results confirmed that, under these dosages, the cell apoptosis was mainly due to photon radiation effect and independent of GO presence. When increasing the photon dose to 6 Gy or 8 Gy, the NPC-BM1 cells without GO showed 15–19% cell apoptosis under the radiation influence [54,55]. At these high radiation dosages (6–8 Gy), the GO-treated sample exhibited increased mortality rate to 26–27%. There was a statistically significant difference in the survival rates (*p*-value < 0.01) between the radiated samples with and without GO exposure. The GO wrapping on the cancer cell surface may reduce X-ray penetration and restrict cell membrane damage [56] at photon radiation doses of up to 5 Gy. When increasing the photon dose to 6 Gy or 8 Gy, the NPC-BM1 cells without GO have 15–19% cell apoptosis whereas the GO-treated samples increased to 26–27%. A higher photon dose (>5 Gy) could break the GO-wrapped layer and allow the photon energy to penetrate the NPC-BM1 cells, thus damaging the surface of the cell membrane by generating free radicals [12]. The conformationally sensitive proteins and/or DNA/RNA within the cells upon GO treatment (as shown in previous section) may become more vulnerable and cause higher cell apoptosis than that without GO exposure. In addition, the energy absorbed by the radiation-stimulated GO could rapidly transfer the level of electrons and consequently induce heat [57] into NPC-BM1 cells. Habiba et al. reported that HT29 tumor cells were radiation-resistant up to 4 Gy when exposed to GO-silver nanoparticles, whereas the survival fraction was reduced by or to 10% at 6 Gy, along with DNA damage. Our work and Habiba et al.’s study confirms the radiosensitization of GO-based nanomaterials [54].

The corresponding cell morphology before and after photon irradiation of GO-treated NPC-BM1 cells is shown in Figure 8. The cell morphology was affected by the radiation, as indicated by the arrow mark, which led to cell damage and apoptosis of the NPC-BM1 cells. Thus, a low GO concentration not only improved the radiation therapy but also reduced the dosage-limiting toxicity. Such GO concentration of 0.1 µg·mL^−1^ could be employed as an active radioprotective agent in therapeutic and occupational pathways whereas the high radiation doses (>5 Gy) induce cellular damage to the NPC-BM1 cells. These findings suggest that nanometric GO is promising as a radioactive agent for radiation therapy.

## 4. Conclusions

In summary, GO nanosheets were exfoliated from graphene using the modified Hummers method. Probe ultrasonication was further applied to reduce the GO size to 100–200 nm. The nanometric GO particles did not induce cytotoxicity on NPC-BM1 cancer cells at tested concentrations. However, the GO-induced conformational changes to proteins and DNA/RNA in cells were illustrated from the SR-FTIRM mapping results. The photon radiation in conjunction with GO treatment increased NPC-BM1 cell apoptosis to 27% when the photon radiation dose was increased to 8 Gy. The findings of the present work demonstrate the potential biological applicability of nanometric GO in different areas, such as targeted drug delivery, cellular imaging, and radiotherapy.

## Figures and Tables

**Figure 1 materials-14-01396-f001:**
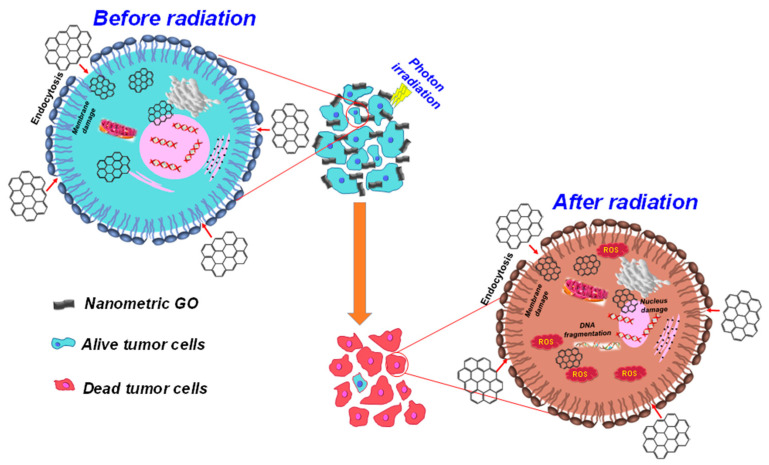
Schematic illustration of GO interaction with cancer cells and the photon irradiation effects of GO.

**Figure 2 materials-14-01396-f002:**
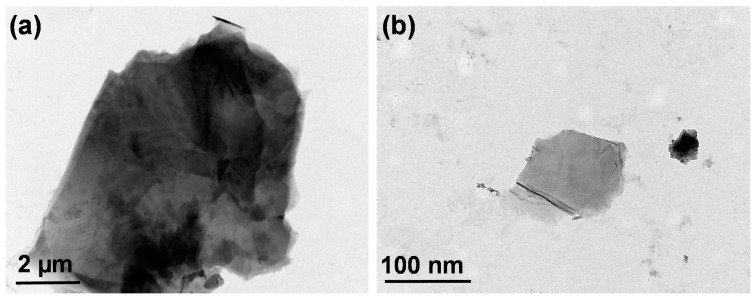
Figure **2.** Transmission electron microscopy images of (**a**) graphite and (**b**) graphene oxide.

**Figure 3 materials-14-01396-f003:**
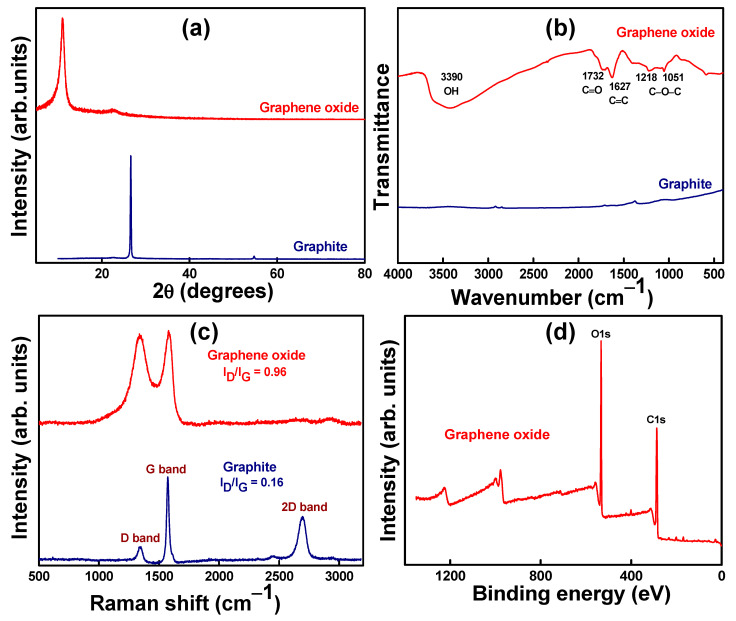
(**a**) X-ray diffraction patterns, (**b**) Fourier transform infrared spectra, (**c**) Raman spectra of graphite and graphene oxide, and (**d**) full scan X-ray photoelectron spectrum of graphene oxide.

**Figure 4 materials-14-01396-f004:**
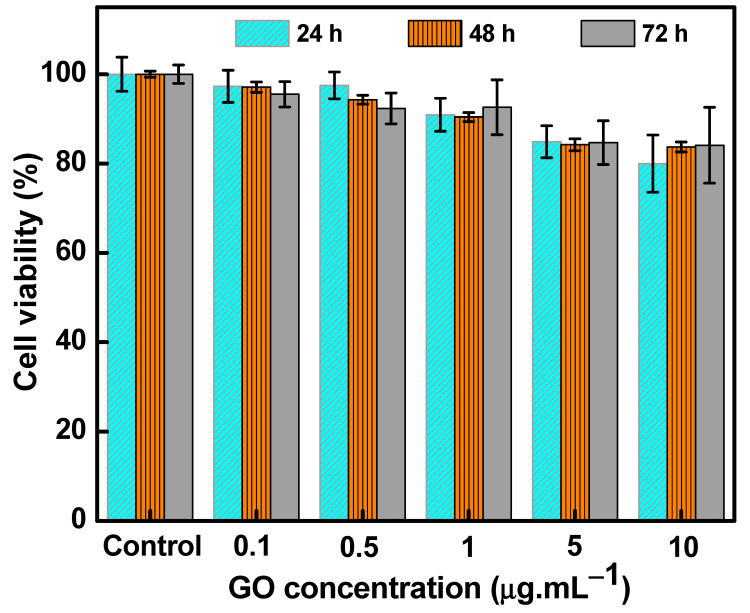
Cell viability of GO-treated NPC-BM1 cells at different GO concentrations at 24, 48, and 72 h. The error bars are standard deviations from triplicate (*n* = 3) measurements.

**Figure 5 materials-14-01396-f005:**
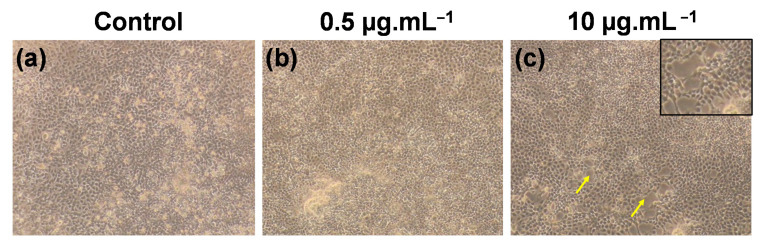
Microscopy images of (**a**) control (without GO exposure) and (**b**,**c**) GO-treated NPC-BM1 cells at 72 h. The GO concentrations are shown above (**b**,**c**). Images were taken at 100× magnification. The arrows indicate the GO interaction with cell and its cell damage.

**Figure 6 materials-14-01396-f006:**
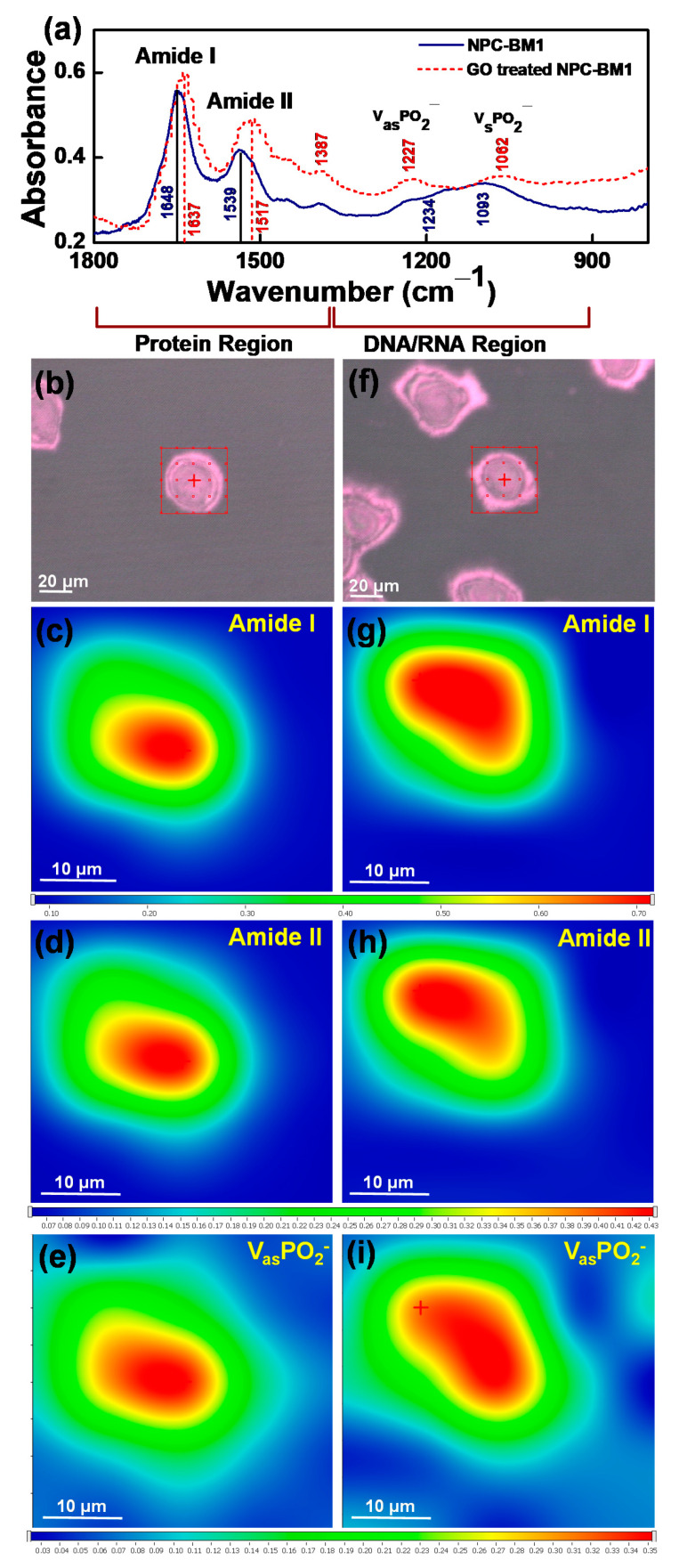
(**a**) FTIR spectrum of pristine and GO-treated NPC-BM1 cells, (**b**,**f**) microscopy images, and SR-FTIRM mapping of amide I, amide II and phosphate groups for (**c**–**e**) pristine NPC-BM1 cell and (**g**–**i**) NPC-BM1 cell treated with GO.

**Figure 7 materials-14-01396-f007:**
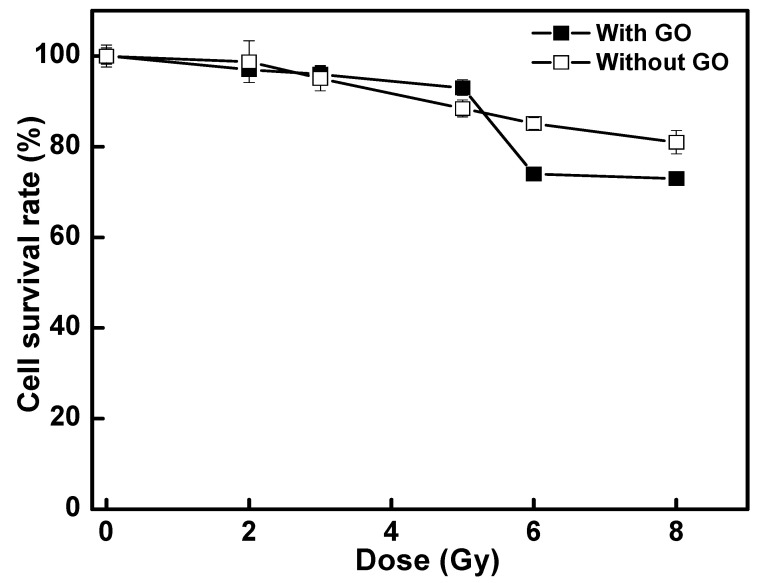
Radiation dose-response curve for NPC-BM1 cancer cells treated without and with a low-concentration of GO (at a concentration of 0.1 µg·mL^−1^). The values are expressed as the mean ± standard deviation of triplicate (*n* = 3) measurements.

**Figure 8 materials-14-01396-f008:**
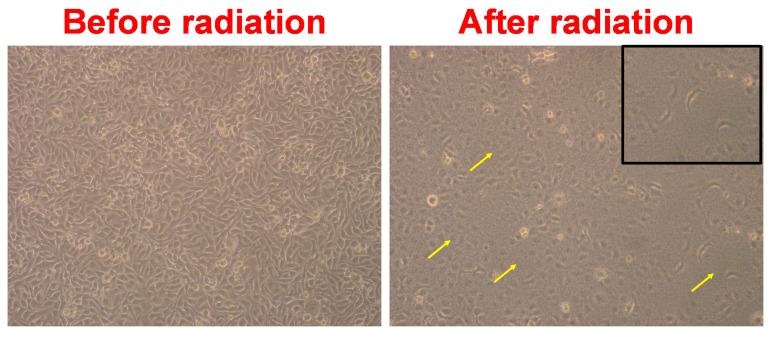
Microscopy images of NPC-BM1 cancer cells before and after photon radiation (8 Gy) at a magnification of 100×. The cancer cells were treated with GO (0.1 µg·mL^−1^). The arrows and the inset image indicate cell damage.

## Data Availability

The data presented in this work are available on request from the corresponding author.

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
