# Peer review of "Graphene Oxide-Induced Protein Conformational Change in Nasopharyngeal Carcinoma Cells: A Joint Research on Cytotoxicity and Photon Therapy"

_materials, 2021, doi:10.3390/ma14061396_

Round 1
Reviewer 1 Report
The manuscript reports the preparation and physical/biological characterization of nanometric GO from graphite. As for biological properties, GO interactions with nasopharyngeal carcinoma cells (NPC-BM1) were studied by synchrotron radiation-based Fourier transform infrared spectroscopy (SR-FTIR). In addition, the effect of different doses of photon radiation on GO treated NPC-BM1 cancer cells was investigated.
Notwithstanding the physical and morphological characterization (FTIR, XRD, RAMAN, TEM and so on) is present in other papers of the authors, the work on cytotoxicity of GO nanosheets on NPC-BM1 cancer cells, in term of altered cellular functions by FTIR mapping, and on the response of these cancer cells to photon irradiation treatment, is very interesting.
The authors can consider the following suggestions:
Pag 3, lines 103-104. The size distribution of GO nanoparticles, showed in figure S1 (supplementary information), was centred at 150-200 nm. The authors state that the final nanoparticle size was in the range 50-100 nm. Does the size shown in figure S1 refer to different GO preparations? If not, how were the 50-100 nm nanoparticles obtained?
Pag 4, lines 132-133. The authors should report the procedure for the preparation of GO suspension (which weight of GO was used for the preparation of the solution to be used in the cell viability tests). Was the GO suspension homogenous and stable?
Pag 5, lines 166-167. Why was the peak at 3390 cm-1 also present in the spectrum of graphite?
Pag 8, line 140. The authors state that “the peak intensity of amide I increased in GO treated cancer cells compared with control group”. With respect to which peak was the intensity increase of the amide I band measured?
Author Response
The manuscript reports the preparation and physical/biological characterization of nanometric GO from graphite. As for biological properties, GO interactions with nasopharyngeal carcinoma cells (NPC-BM1) were studied by synchrotron radiation-based Fourier transform infrared spectroscopy (SR-FTIR). In addition, the effect of different doses of photon radiation on GO treated NPC-BM1 cancer cells was investigated.
Notwithstanding the physical and morphological characterization (FTIR, XRD, RAMAN, TEM and so on) is present in other papers of the authors, the work on cytotoxicity of GO nanosheets on NPC-BM1 cancer cells, in term of altered cellular functions by FTIR mapping, and on the response of these cancer cells to photon irradiation treatment, is very interesting.
Response: Thank you very much for your high-quality positive commends. We are highly appreciative to you for pointing out our typos and other errors. We have revised the manuscript carefully according to your comments.
The authors can consider the following suggestions:
Q1: Page 3, lines 103-104. The size distribution of GO nanoparticles, showed in figure S1 (supplementary information), was centred at 150-200 nm. The authors state that the final nanoparticle size was in the range 50-100 nm. Does the size shown in figure S1 refer to different GO preparations? If not, how were the 50-100 nm nanoparticles obtained?
Response: Thank you for pointing out this. The prepared GO particle size is ~100-200 nm based on TEM and particle size distribution analysis. The typo error is corrected in the experimental section 2.1, p.3 and also corrected in the entire revised manuscript.
Q2: Pag 4, lines 132-133. The authors should report the procedure for the preparation of GO suspension (which weight of GO was used for the preparation of the solution to be used in the cell viability tests). Was the GO suspension homogenous and stable?
Response: Thank you for pointing out this. Ten milligram of prepared nanometric GO was dissolved in 100 mL DI water to obtain homegeneous GO suspension by ultrasonication to form a stock solution. Then, the GO solution is further diluted in different concentrations (0.1-10 µg mL-1, which were diluted 1, 5, 10, 50 and 100 µL mL-1 from GO stock solution) to investigate the effect on cell viability. The details are included in the Section 2.1 & 2.4, pp. 3-4.
The above figure shows the homogeneous nanometric GO suspension in DI water that was stored for more than three months. No particle sedimentation after prolong time and this confirms that the nanometric GO solution has good stability (and similar with other literature [Ljam et al., International Journal of Heat and Mass Transfer 87 (2015) 92-103]).
Q3: Page 5, lines 166-167. Why was the peak at 3390 cm-1 also present in the spectrum of graphite?
Response: Thank you for pointing out this. The hydroxyl peak at 3390 cm-1 was due to presence of moisture in the sample. Thereby, we reanalyzed the FTIR spectra of raw graphite and no significant peaks are observed. The revised FTIR graph of graphite is included in Fig. 3b and its supporting references are included in 3rd paragraph of section 3.1, p. 5.
Q4: Page 8, line 140. The authors state that “the peak intensity of amide I increased in GO treated cancer cells compared with control group”. With respect to which peak was the intensity increase of the amide I band measured?
Response: The peak at 1644 cm-1 represents the amide I band [Holman et al. Biomolecules 57(6) (2000) 329-335; Mihoubi et al. PLoS One 12 (2017) e0180680] and its intensity increased as compared with control group (without GO treated NPC-BM1 cells) due to change in protein content [Wang et al. Acta Pharmaceutical Sinica B 5(3) (2015) 270-276]. The FTIR spectra of pristine and GO treated NPC-BM1 cells (Fig. 6a) are combined for comparing the peak intensity and wavenumber shift. In addition, the SR-FTIRM mapping images of amide II and phosphate are included in Fig. 6. The detail is included in the section 3.3, pp. 8-10.

Reviewer 2 Report
The paper entitled “Examining the nanometric graphene oxide treated nasopharyngeal carcinoma cells and FTIR microspectroscopy using synchrotron radiation” that you kindly submitted for publication in the “Materials” now been considered.
In the manuscript, the authors compared the difference between graphite and graphene oxide (GO) by showing four data such as X-ray diffraction patterns, IR spectra, Raman spectra, and full scan X-ray photoelectron spectrum. To show the effect of GO on NPC-BM1 cells, the authors examined cell viability, FTIR spectrum, and so on. The GO might be promising agent on cancer therapeutic strategy especially in radiotherapy, which have been being studied by many groups in many cancers. Here, the authors have logical and rational approach and tried to prove, however, there are a few things to be shown before publication.
Specific comments:
- NPC-BM1 treated with GO showed dose-dependent damage in terms of MTT in figure 4 and 5, which was further analyzed by FTIR spectrum in Figure 6.
Then, the authors showed the radiation dose response for NPC-BM1 with a low-concentration of GO. Please show the radiation dose response for NPC-CM1 with or without GO in one figure and address if it is beneficial for radiation therapy to use GO in NPC-BM1 cells
- In Figure 6, please show statistical analyses of the experiments for better understanding.
Author Response
The paper entitled “Examining the nanometric graphene oxide treated nasopharyngeal carcinoma cells and FTIR microspectroscopy using synchrotron radiation” that you kindly submitted for publication in the “Materials” now been considered.
In the manuscript, the authors compared the difference between graphite and graphene oxide (GO) by showing four data such as X-ray diffraction patterns, IR spectra, Raman spectra, and full scan X-ray photoelectron spectrum. To show the effect of GO on NPC-BM1 cells, the authors examined cell viability, FTIR spectrum, and so on. The GO might be promising agent on cancer therapeutic strategy especially in radiotherapy, which have been being studied by many groups in many cancers. Here, the authors have logical and rational approach and tried to prove, however, there are a few things to be shown before publication.
Response: Thank you very much for your high-quality positive commends. We have revised the manuscript carefully according to your comments.
Specific comments:
Q1: NPC-BM1 treated with GO showed dose-dependent damage in terms of MTT in figure 4 and 5, which was further analyzed by FTIR spectrum in Figure 6. Then, the authors showed the radiation dose response for NPC-BM1 with a low-concentration of GO. Please show the radiation dose response for NPC-BM1 with or without GO in one figure and address if it is beneficial for radiation therapy to use GO in NPC-BM1 cells.
Response: Thanks for your insightful comment. The graph of dose-dependent radiation effect on NPC-BM1 cells without GO is added in Fig. 7 and its discussion is included in Section 3.4, p. 9.
Q2: In Figure 6, please show statistical analyses of the experiments for better understanding.
Response: Thank you for pointing out this. The statistical analysis of microscopic images and the equivalent diameters of the different regions of SR-FTIRM (Fig. 6) was included in the revised manuscript, pp. 8-9.
Round 2
Reviewer 2 Report
The responses to my comments were acceptable. the revised manuscript was much improved.